# The Clinical Relevance of Distinguishing Between Simple and Complex Adnexal Cystic Structures by Ultrasound in Peri- and Postmenopause

**DOI:** 10.3390/cancers17081370

**Published:** 2025-04-20

**Authors:** Balazs Erdodi, Gergo Jozsef Szollosi, Zoltan Toth, Zoard Tibor Krasznai, Attila Jakab

**Affiliations:** 1Department of Obstetrics and Gynecology, Faculty of Medicine, University of Debrecen, 4032 Debrecen, Hungary; ztoth@med.unideb.hu (Z.T.); krasznai.zoard@med.unideb.hu (Z.T.K.); ja@med.unideb.hu (A.J.); 2Doctoral School of Clinical Sciences, University of Debrecen, 4032 Debrecen, Hungary; 3Coordination Center for Research in Social Sciences, Faculty of Economics and Business, University of Debrecen, 4032 Debrecen, Hungary; szollosi.gergo@etk.unideb.hu

**Keywords:** ovary, ovarian cancer, ultrasonography, complex cyst, simple cyst, CA125, diagnostics, basic ultrasound scan

## Abstract

Cystic ovarian structures affect women of all ages. As a result of the IOTA studies, ultrasound has become the first diagnostic choice for pelvic masses, regardless of patient age and reproductive status. Although mathematical models are highly accurate, their application depends on standardized ultrasound assessments, which remain inconsistently implemented in many clinical settings. This study aimed to investigate the use of simple ultrasonographic features adjusted to reproductive status to distinguish benign from malignant masses. Size and complex morphology in perimenopause can be good markers for selecting patients for further preoperative investigation.

## 1. Introduction

The prevalence of cystic ovarian structures is increased in peri- and postmenopause. Bimanual examination, serum tumor markers, and transvaginal ultrasound examinations can also help to detect adnexal masses. As the majority of these cystic structures show no malignant potential, it is essential to easily and effectively select high-risk masses for appropriate care [1]. Functional and malignant cysts can be characterized by their symptoms at the same time. Therefore, more than 60% of ovarian cancers are detected at stage III or IV [2]. In case of ovarian malignancies, CA-125 is recommended to increase detection rate, but it can be used effectively together with transvaginal US scans [3,4,5]. Iatrakis et al. proposed a New Risk Malignancy Index (NRMI) to enhance the prediction of ovarian cancer in women presenting with pelvic masses. This bicentric retrospective study compared the NRMI to the classical Risk Malignancy Index (RMI) by incorporating not only CA-125 levels and ultrasound findings, but also known risk (e.g., infertility, endometriosis, and PCOS) and protective factors (e.g., pregnancy and breastfeeding) [6].

Since transvaginal probes have been introduced, several scoring systems have been developed to increase the detection rate of grayscale ultrasonography [7,8,9,10,11]. Papillary projections, thick and irregular walls, and septated or large multilocular cysts have been proven to carry a higher risk of malignancy, as they are called complex cysts [12,13,14].

Pavlik et al. analyzed over 20,000 women undergoing serial transvaginal ultrasonography and documented the natural course of ovarian abnormalities. They reported that most unilocular cysts were benign and resolved spontaneously, with a low risk of malignancy, even when monitored over extended periods. The study highlighted that morphology and growth behavior were stronger predictors of malignancy than size alone. These findings support conservative management for most asymptomatic ovarian cysts, emphasizing the importance of sonographic follow-up over immediate surgical intervention [15].

As a result of international multicenter studies, evidence-based scoring systems have been developed by the IOTA (International Ovarian Tumour Analysis) group [11,16,17]. The work of this collaboration group has been dominant in the ultrasound-based diagnosis of adnexal masses over the last two decades and has raised the diagnostic process to an evidence-based and reliable level. Since the development of the ADNEX model, a mathematical model is on hand for everyone, with different ways to distinguish between benign and malignant lesions and an efficacy reaching expert-level opinion, so-called ‘pattern recognition’ [18,19,20].

Risk factors affecting the formation of cystic ovarian structures, such as genetic predisposition, multiparity, and previous gynecological surgeries, have been examined in several studies [21,22].

The aim of this study was to determine the efficacy of simple grayscale ultrasound markers and the additive value of CA-125 in the detection and triage of ovarian cystic structures in peri- and postmenopause.

## 2. Materials and Methods

This study was performed in the Department of Obstetrics and Gynecology at the University of Debrecen, Debrecen, Hungary. The imaging properties of peri- (PEM) and postmenopausal (POM) ovarian cysts were examined preoperatively. All patients underwent transvaginal ultrasound examinations preoperatively (ATL HDI-3000, Bothell, DC, USA, equipped with 5.9 MHz transvaginal probe and Medison Accuvix XQ, Medison Co., Ltd., Seoul, Republic of Korea, equipped with a 5–8 MHz transvaginal probe Kretztechnik AG, Zipf, Austria). Patients over the age of 40 with cystic adnexal masses were consecutively recruited. Patients without clinical signs of menopause or climacteric symptoms were categorized into the PEM group, as well as those who previously underwent hysterectomy under the age of 50 (range: 40–54 years, average: 45.57 years). A lack of regular periods for more than one year or hysterectomy in patients over 50 years of age were selected for the POM group (range: 41–88 years, average: 61.24 years). Overall, 343 patients with 379 cystic structures were involved in the study. US examinations were performed independent of the menstrual cycle and were repeated within three months in 168 cases (44.32%). When follow-up was performed, only the preoperative US finding was chosen for inclusion. According to the US findings, the following two groups were formed: (1) simple cysts: unilocular, anechoic cysts without any solid structure and (2) complex cysts: cystic structures with different parameters and even containing solid parts. Examples shown in Figure 1. Imaging characteristics and size were matched with histology and CA125 levels. The cut-off level for CA-125 was 35 kIU/L. According to the size of the cysts, two subgroups were formed within each reproductive group, and 5 cm was chosen for this purpose. Risk factors affecting the formation of cystic structures in the ovary were also observed in this study, such as parity, previous pelvic surgeries, and family history of ovarian cancer.

Statistical analysis was performed using SPSS 10.0. Significance was checked using the Mann–Whitney, Chi-square, and Kruskal–Wallis tests.

## 3. Results

Within the perimenopausal (PEM) cohort, 75 simple cysts were identified—comprising 32 lesions with diameters less than 5 cm and 43 lesions measuring 5 cm or more—as well as 122 complex cysts (29 < 5 cm and 93 ≥ 5 cm). In contrast, the postmenopausal (POM) group exhibited 49 simple cysts (9 < 5 cm and 40 ≥ 5 cm) and 135 complex cysts (15 < 5 cm and 120 ≥ 5 cm). Notably, in the PEM group, malignancy was confined exclusively to complex cysts larger than 5 cm (n = 16, 17.58%), whereas the POM group included 40 malignant cases, 3 of which were under 5 cm. Furthermore, the majority of the PEM cysts were functional (54.36%), while in the POM group, serous cysts predominated (38.04%), followed by malignant (21.74%) and mucinous cysts (13.04%). It is also important to note that functional cysts were detected in 5.43% of the postmenopausal cases (n = 10). Comprehensive details are provided in Table 1.

CA 125 determination was performed in 194 cases (51.19%). In the case of masses larger than 5 cm, the distribution of CA125 values was different between simple and complex structures (*p* = 0.003). Postmenopausal status was responsible for this significance, as we divided the patients into PEM and POM groups (PEM: *p* = 0.112, NS while POM: *p* = 0.009). There was no significant difference in CA125 values between the PEM and POM groups in case of masses smaller than 5 cm (*p* = 0.51). On the other hand, a significant correlation was also detected between histological findings and abnormal biomarker values in the POM group (*p* < 0.05), while the same could not be seen in the PEM group (*p* = 0.159). Naturally, higher serum biomarker levels were detected in borderline and malignant tumors (*p* < 0.001). Moreover, the distribution of CA125 values was different in complex-morphology-associated histological results (*p* < 0.001), but not in the simple morphological group (*p* = 0.171).

The occurrence of malignant or borderline lesions was significantly higher in the POM group (*p* = 0.001) and also among complex masses (*p* < 0.001). Complex morphology carried a significant risk for malignancy in both perimenopausal and postmenopausal adnexal masses (*p* = 0.005). The same tendency was not noticed in lesions with simple ultrasonographic features (*p* = 0.402). The largest diameter of the mass affected the histological findings. In the whole study population, borderline and malignant masses were significantly larger (*p* < 0.001). This difference was supported by perimenopausal cases (*p* < 0.001), while the same was not noticed in the case of postmenopausal patients (*p* = 0.147). Details are shown in Figure 2 and Figure 3.

The significance of the 5 cm cut-off is underscored by the presence of a statistically significant difference (*p* < 0.001) in the largest tumor diameters among complex cysts with differing histological outcomes—a correlation that was not observed among simple cysts (*p* = 0.109). Independent of this size threshold, histological variability remained statistically significant within complex morphological categories (*p* < 0.001), and, to a lesser extent, also within simple morphological groups (*p* = 0.017).

Several factors influencing the development of ovarian cystic structures are summarized in Table 2. Notably, 84.96% of cases occurred in women with a history of at least one delivery. Among the total cohort, 210 cysts (55.41%) were associated with patient-reported symptoms, with pelvic pain being the most frequent, reported in 160 cases (42.22%). A statistically significant positive correlation was identified between the number of deliveries and the presence of pelvic pain, indicating that women with two or more deliveries were more likely to report this symptom (*p* = 0.046). Conversely, no significant association was found between tumor size and the presence of clinical symptoms (*p* = 0.842), Figure 4. Of the 379 cases, laparotomy was performed in 336 (88.65%). A history of previous pelvic surgery was documented in 114 cases (30.08%), with hysterectomy being the most common (n = 71, 18.73%). A family history of ovarian cancer was recorded in only four cases (1.06%).

## 4. Discussion

Cystic structures of the ovaries can be found in 11–16% of postmenopausal cases and more frequently perimenopausal cases [23]. The risk of the malignant transformation of benign cysts is still an open question [24]. It has been proven in several studies that the risk of malignant transformation is directly proportional to the number of EGF receptors in the cystic fluid [25,26,27]. In our study, this connection was not examined.

According to the fact that ovarian pathologies depend on the functional state of the ovaries, the division of the population into peri- and postmenopausal groups was necessary. The menopausal status of a patient is involved in a few scoring systems, among which the most widely used is the Risk of Malignancy Index (RMI) [28]. The New RMI was introduced by Iatrakis et al. in 2018 and demonstrated a superior diagnostic sensitivity, identifying malignancy in 66.7% of cases, compared to 29.6% with the original RMI. Despite promising results, the authors emphasized the need for validation in broader populations before clinical implementation, acknowledging potential overfitting due to the limited sample size (N = 27) [6].

Koonings et al. found a 13% risk for malignant transformation in perimenopause, which increased to 45% in the postmenopausal group [29]. The likelihood of malignant transformation increases with cyst size. Modesitt et al. advised a 10 cm cut-off level, while Osmers et al. found a diameter of 3 cm for the border between low-risk and high-risk groups [30,31]. Auslender et al. and Reimer et al. recommended a cut-off of 5 cm, and this was used in this study as well [25,32]. The IOTA group advises several diagnostic algorithms for the detection of adnexal masses. In these algorithms, most ultrasound characteristics, such as tumor size, are independent of menopausal status. On the other hand, tumor size affects the performance of subjective assessment, mathematical models (LR1 and LR2), the IOTA simple rules, and also the Risk of Malignancy Index (RMI) in correctly discriminating between benign and malignant adnexal masses [33]. Using the IOTA simple rules to determine reproductive status is not a basic step. Subjective assessment, commonly referred to as ‘pattern recognition’, has demonstrated a superior diagnostic performance when performed by experienced sonographers [17]. On the other hand, the most developed mathematical model, the so-called ADNEX model, uses not just the largest diameter of the lesion, but also the largest diameter of the presenting solid component, if there is any [18]. Based on a recent paper by Landolfo et al., the suggested method for the routine management of adnexal masses is the ‘two-step strategy’ using the benign descriptors (BDs) and the ADNEX model as a second step in unclassifiable cases. According to their data, 37% of adnexal lesions can be treated as benign just using the benign descriptors, which use 10 cm as a threshold for discrimination. Unclassifiable masses were analyzed using the ADNEX model. An excellent diagnostic performance was reported with the use of this strategy, providing an AUC of 0.94. Discrimination between benign and malignant masses was better calibrated in postmenopausal patients, but the diagnostic efficiency was nearly the same in both groups [34].

In the present study, the prevalence of malignancy among perimenopausal cystic lesions was found to be 4.22%, which is consistent with previously reported rates by Ekerhovd et al. (0.7%) and Osmers et al. (0.8%) [2,35]. Sonographic features such as echogenicity, wall characteristics, septations, papillary projections, and the presence of solid components are critical in assessing cyst complexity and, consequently, the risk of malignancy. These parameters are essential for accurate diagnostic decision making. Although multiple scoring systems have been developed by researchers such as Timmerman and Ueland—focusing on cyst volume and wall morphology—none of these, including the IOTA mathematical models, were applied in the current analysis [9,11].

Among postmenopausal women, the risk of malignancy is notably higher, a finding supported by our data, showing 40 malignant cysts (10.55%), of which 97.5% (n = 39) demonstrated complex sonographic morphology. This aligns with the findings of Osmers et al. (9.6%) and Ekerhovd et al. (10%) [2,31], reinforcing the recommendation for surgical intervention in such cases. Conversely, simple cysts in postmenopausal patients were associated with malignancy in only 0.26% of cases, suggesting that routine surveillance, with or without CA-125 assessment, may be an appropriate management strategy. The benign nature of most unilocular ovarian cysts and the safety of serial sonographic follow-up have been confirmed in large prospective studies, such as that by Pavlik et al., involving over 20,000 women [15]. Our findings support the conservative management of simple adnexal cysts in asymptomatic postmenopausal women, consistent with the Society of Radiologists in Ultrasound consensus guidelines, which recommend no follow-up for cysts of ≤1 cm and annual surveillance for larger but morphologically benign lesions [36]. Our results align with previous research indicating that septated cysts resolve quicker than unilocular cysts, suggesting that cyst morphology significantly influences the natural history of ovarian cysts [37]. A recent review by Liu et al. reported a malignancy rate of 1 in 10,000 for simple postmenopausal cysts, further validating this conservative approach [38].

CA-125, as a standalone biomarker, demonstrates a limited efficacy in early-stage ovarian cancer detection, with a sensitivity of approximately 60% and a specificity of 99% [39,40]. When combined with transvaginal ultrasonography, its sensitivity improves to as much as 85% [28]. In the current study, the overall detection rate based on elevated CA-125 levels was 30.41%. According to our findings in postmenopause in the case of complex cysts larger than 5 cm, an elevated CA-125 level may give added value to the suspicion of malignancy. Otherwise, in perimenopause and in the case of postmenopausal simple cysts, the prevalence of an elevated CA-125 level did not differ between benign and malignant cases. Notably, CA-125 concentrations can be elevated in a variety of benign conditions, including endometriosis, pelvic inflammatory disease, uterine fibroids, and Meigs syndrome [41,42]. Consequently, numerous studies support the combined use of CA-125 and ultrasound to enhance diagnostic accuracy, particularly in perimenopausal patients, and caution against relying on CA-125 levels alone for malignancy risk assessment [19,43,44,45]. Moreover, it has been demonstrated that elevated CA-125 values do not significantly improve the distinction between benign and malignant lesions, especially when assessments are performed by experienced sonographers [46]. Nonetheless, CA-125 remains an integral component of advanced diagnostic models such as the ADNEX model, where it contributes to the differentiation of malignant subtypes [18].

The implementation of standardized risk stratification systems, such as the O-RADS US risk score, has demonstrated potential in reducing unnecessary surgical interventions. Shen et al. reported that 42% of surgically resected lesions could have been managed conservatively if the O-RADS criteria had been applied, highlighting the system’s effectiveness in distinguishing between benign and malignant adnexal masses [47]. Recent large-scale data support limiting surveillance duration for stable adnexal masses beyond one year, given the negligible risk of malignancy observed beyond this interval [48].

## 5. Conclusions

A simple diagnostic algorithm based on the simple or complex morphology of ovarian masses during ultrasonography can easily decrease the number of surgeries and high hospital costs. Categorizing cystic structures into these two groups can be a reliable alternative to mathematical models in the case of simple cysts in both peri- and postmenopause. Regarding mass size, it is also recommended to use 5 cm as a threshold for categorizing high-risk cases for further surgical intervention, with the implementation of CA125 in the case of postmenopausal patients with complex cyst morphology.

## Figures and Tables

**Figure 1 cancers-17-01370-f001:**
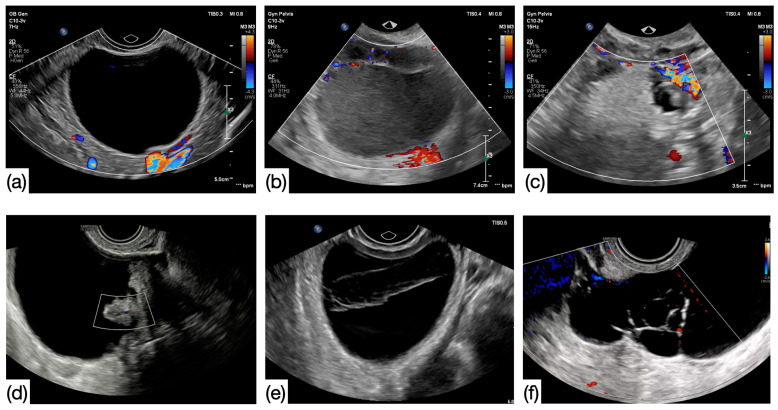
Ultrasound images of simple (**a**) and complex cysts (**b**–**f**). (**a**) Simple cyst: unilocular, anechoic, smooth, and regular-walled cyst without any solid component (**b**) ground glass content—endometrioma, (**c**) mixed echogenicity content—dermoid, (**d**) solid papillary projection, (**e**) hemorrhagic content, and (**f**) multilocular cyst (images of B.E.).

**Figure 2 cancers-17-01370-f002:**
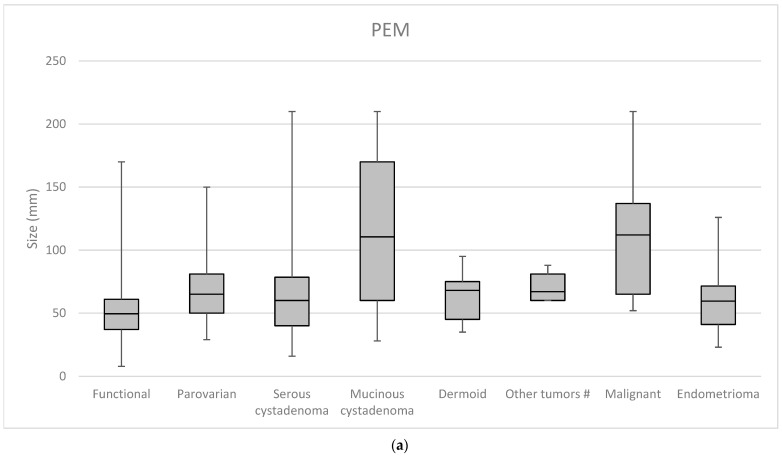
Correlation between tumor size and histological finding: (**a**) perimenopausal (PEM) patients and (**b**) postmenopausal (POM) patients. # Other tumors: Fibroid, Brenner tumor, Struma ovarii, Hydrosalpinx.

**Figure 3 cancers-17-01370-f003:**
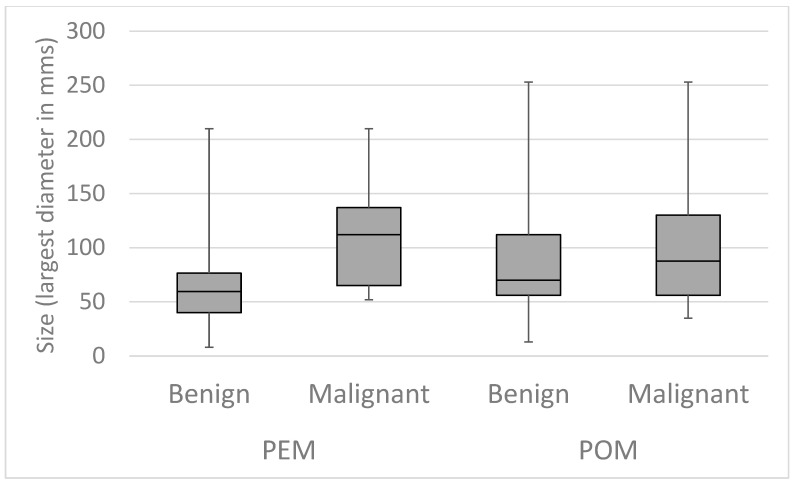
Link between histology and the largest diameter of adnexal masses according to menopausal status: perimenopausal (PEM) malignant tumors are significantly larger than benign masses; no significant difference can be observed in the size of postmenopausal (POM) benign and malignant lesions.

**Figure 4 cancers-17-01370-f004:**
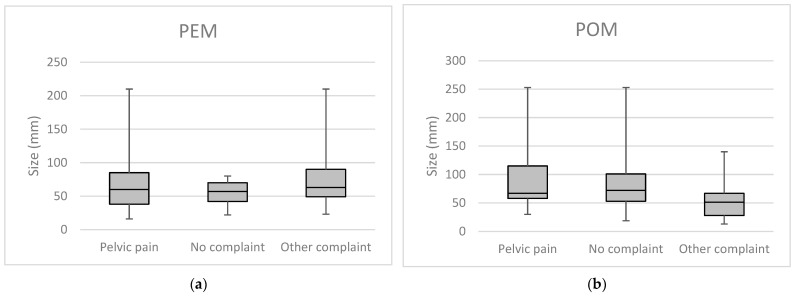
Correlation between tumor size and patient complaints. (**a**) Perimenopausal (PEM) patients complaints and their correlation with tumor size and (**b**) postmenopausal (POM) patient complaints and their correlation with tumor size.

**Table 1 cancers-17-01370-t001:** Correlation between cyst size, ultrasound morphology, and histology. N = 379 (100%).

	PEM Simplex		PEM Complex		POM Simplex		POM Complex	
	<5 cm	≥5 cm	<5 cm	≥5 cm	<5 cm	≥5 cm	<5 cm	≥5 cm
Functional cysts								
Corp. lut. cyst. hem.	8 (25.00%)	5 (12.20%)	6 (20.69%)	12 (12.90%)			2 (13.33%)	5 (4.17%)
Follicular cyst	5 (15.63%)	6 (14.63%)	6 (20.69%)	2 (2.15%)	1 (11.11%)			2 (1.67%)
Parovarian cysts	1 (3.13%)	11 (26.83%)	3 (10.34%)	3 (3.23%)	2 (22.22%)	5 (12.50%)	3 (20.00%)	5 (4.17%)
Benign								
Cystadenoma serosum	16 (50.00%)	16 (39.02%)	3 (10.34%)	21 (22.58%)	5 (55.56%)	27 (67.50%)	4 (26.67%)	34 (28.33%)
Cystadenoma mucinosum	1 (3.13%)	2 (4.88%)	1 (3.45%)	10 (10.75%)		4 (10.00%)	2 (13.33%)	18 (15.00%)
Dermoid			4 (13.79%)	10 (10.75%)				5 (4.17%)
Endometriosis	1 (3.13%)	1 (2.44%)	6 (20.69%)	12 (12.90%)				3 (2.50%)
Fibroid				3 (3.23%)		2 (5.00%)	1 (6.67%)	7 (5.83%)
Hydrosalpinx					1 (11.11%)	1 (2.50%)		2 (1.67%)
Struma ovarii				1 (1.08%)				1 (0.83%)
Brenner tu.				3 (3.23%)				2 (1.67%)
Malignant				16 (17.20%)		1 (2.50%)	3 (20.00%)	36 (30.00%)

**Table 2 cancers-17-01370-t002:** Risk factors affecting the formation of ovarian cysts.

Risk Factors	PEM Simplex		PEM Complex		POM Simplex		POM Complex		Total (% of Total Cases)
	<5 cm	≥5 cm	<5 cm	≥5 cm	<5 cm	≥5 cm	<5 cm	≥5 cm	
Parity									
Nulliparous	2 (6%)	10 (24%)	6 (21%)	14 (15%)	1 (11%)	1 (3%)	3 (20%)	20 (17%)	57 (15.04%)
1–2 children	20 (63%)	23 (56%)	19 (66%)	65 (70%)	6 (67%)	31 (78%)	8 (53%)	78 (65%)	250 (65.96%)
3–4 children	10 (31%)	8 (20%)	3 (10%)	13 (14%)	2 (22%)	7 (18%)	4 (27%)	20 (17%)	67 (17.68%)
≥5 children			1 (3%)	1 (1%)		1 (3%)		2 (2%)	5 (1.32%)
Family history of ovarian cancer	2 (6%)		1 (3%)	1 (1%)					4 (1.06%)
Previous pelvic surgery									
Hysterectomy	5 (16%)	4 (10%)	3 (10%)	17 (18%)	3 (33%)	2 (5%)	4 (27%)	33 (28%)	71 (18.73%)
Adnexectomy	1 (3%)	1 (2%)	1 (3%)	8 (9%)			1 (7%)	2 (2%)	14 (3.69%)
Laparotomy	1 (3%)	4 (10%)	1 (3%)	10 (11%)		1 (3%)		5 (4%)	22 (5.80%)
Punction		3 (7%)	1 (3%)	1 (1%)		1 (3%)	1 (7%)		7 (1.85%)
No previous surgery	25 (78%)	29 (71%)	23 (79%)	57 (61%)	6 (67%)	36 (90%)	9 (60%)	80 (67%)	265 (69.92%)
Total (n)	32	41	29	93	9	40	15	120	379 (100%)

## Data Availability

In accordance with the journal’s guidelines, the data presented in this study are available upon request from the corresponding author for the reproducibility of this study if such is requested.

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
