# Peer review of "The Clinical Relevance of Distinguishing Between Simple and Complex Adnexal Cystic Structures by Ultrasound in Peri- and Postmenopause"

_cancers, 2025, doi:10.3390/cancers17081370_

Round 1
Reviewer 1 Report
Comments and Suggestions for Authors
• The main question addressed by the research is the determination of the reliability of simple ultrasound markers and CA-125 measurement in diagnosing peri- and postmenopausal ovarian masses.
• The topic is relevant to the field without addressing a major specific gap.
• Compared with other published material, it adds to the subject area the suggestion of surgery in all complex cysts IRRESPECTIVELY of the size.
• Regarding the methodology, no further improvements are necessary.
• The conclusions are consistent with the evidence and arguments presented and they address the main question posed, concluding that CA-125 does not give added value to the detection of malignancy, although this is not in agreement with other published data.
• The references are appropriate. However, references in very close relation to the subject are missing. Example: J BUON 2018 Sep-Oct;23(5):1380-1383. A new risk malignancy index to predict ovarian cancer: a bicentric preliminary study.
Reviewer 2 Report
Comments and Suggestions for Authors
Its a welcome article about the attitude towards adnexal cysts,as surgery is practiced in exces due to relative indications,and a well structured article.Introduction provides the necessary data regarding the current equivocal attitude about adnexal cysts and attempts to assess diagnostic and treatment algorithms.Materials and methods is adequate described ,well structured and exposed,with informațional about ultrasound features of the cysts for entire lot of pacients.Results are clearly presented with well constructed graphs and tables that show the correct picture of the study in question.Disscussions are coherent but in the first paragraph it is stated that the direct proportionality between the risk of malignant transformation and the number of EGF receptors has not been found,but there is no mention in the study about this.In the rest,the results are compatible with the data of the cited studies,that emphasize the high risk of malignant transformation for complex cysts in POM with size over 5cm.Conclusions are logical,by common sense and to thepoint.The English Language uzez is fluent,and referrences are well chosen and correctly cited.
Reviewer 3 Report
Comments and Suggestions for Authors
Background and results have been properly presented with clear conclusione well supported by results. I didn't find the manuscript original and with a relevant impact on everyday clinical practice.
Reviewer 4 Report
Comments and Suggestions for Authors
I think that although well written, this manuscript should not be accepted for publication because it adds nothing to the existing literature. Multiple studies, also cited by the authors, have shown similar results or better tools to be able to discriminate adnexal masses.
Reviewer 5 Report
Comments and Suggestions for Authors
Although the authors aimed to evaluate the efficacy of simple grayscale ultrasound markers and the additional value of CA-125 in detecting and triaging ovarian cystic lesions in peri- and postmenopausal women, the study fails to establish the clinical utility of ultrasound. While the results indicate that preoperative ultrasound findings differ between the peri- and postmenopausal periods—such as the observation that functional cysts are common during perimenopause and that malignancy was found only in cysts with complex morphology larger than 5 cm—the study does not propose objective criteria, such as a specific size cutoff, that could inform clinical decision-making. Ultimately, the study appears to be a summary of routine clinical data and does not present any novel or original findings.
Reviewer 6 Report
Comments and Suggestions for Authors
The aim of this study was to determine the efficacy of simple greyscale ultrasound markers and the additive value of CA-125 in the detection and triage of ovarian cystic structures in peri- and postmenopausal women. Ovarian cysts are common in postmenopausal women. The exact prevalence is unknown given the limited amount of published data and the lack of established screening programmes for ovarian cancer. This is an interesting topic, and it focuses on an important and timely issue.
After reading this paper several major issues came to my mine.
Results section – Representative photographs of functional cysts, both simple and complex, and malignant cysts should be included in the results section. These data would benefit from being presented in summary figures.
Discussion section – Authors wrote that “ It was proven in several studies that the risk of malignant transformation is directly proportional to the number of EGF receptors in the cystic fluid [23-25]. In our study, we did not observe this connection.” However, I cannot find EGFR analysis in results. For this reason, this sentence should be deleted.
Round 2
Reviewer 3 Report
Comments and Suggestions for Authors
I appreciated this new version of the manuscript with relevant references and better and properly presented resultts with adequate methods.

Reviewer 5 Report
Comments and Suggestions for Authors
I made the following comments on the first edition.----
The authors aimed to evaluate the effectiveness of simple grayscale ultrasound markers and the additional value of CA-125 in the detection and classification of ovarian cystic lesions in women around the time of menopause, but this study did not establish the clinical usefulness of ultrasound. The results show that the findings of preoperative ultrasound examinations differ between the premenopausal and postmenopausal periods. For example, the observation that functional cysts are more common in the peri-menopausal period, and malignant tumors are only seen in cysts with complex morphology that exceed 5 cm in size. However, this study does not propose any objective criteria, such as a specific size cutoff value, that could be used to aid clinical decision-making. Ultimately, this study is a compilation of routine clinical data, and does not present any new or unique findings.-----
I have read the revised paper that you have submitted this time. I recognize that you have revised the expression of the results and the addition of the discussion section. However, I did not find any changes in the concept and conclusion of the study. Therefore, I could not change my overall evaluation.
